# Empowering Maternal Choice: Exploring Factors Influencing Early Postpartum Contraceptive Adoption Intention Among Pregnant Women in Northeast Ethiopia

**DOI:** 10.3390/ijerph21111418

**Published:** 2024-10-25

**Authors:** Niguss Cherie, Muluemebet Abera Wordofa, Gurmesa Tura Debelew

**Affiliations:** 1Population and Family Health Department, Faculty of Public Health, Institute of Health, Jimma University, Jimma P.O. Box 5195, Ethiopia; mulu_abera.ts2009@yahoo.com (M.A.W.); gurmesatura@gmail.com (G.T.D.); 2Reproductive and Family Health Department, School of Public Health, College of Medicine and Health Sciences, Wollo University, Dessie P.O. Box 1145, Ethiopia

**Keywords:** intention, early postpartum contraception, pregnant women, Dessie, Kombolcha

## Abstract

Background: Despite progress in access to family planning services in many sub-Saharan African countries in recent decades, advances in early postpartum contraceptive adoption remain low, and the unmet need for early postpartum contraceptives is high. According to the Ethiopia Demographic and Health Survey report, early postpartum modern contraceptive method uptake is still unacceptably low in Ethiopia. Objectives: This study aimed to determine the magnitude of intention to adopt early postpartum modern contraceptive methods and its associated factors among pregnant women in Dessie and Kombolcha town zones, northeast Ethiopia. Methods: A community-based cross-sectional study was deployed from 15 January–15 February 2023, in the Dessie and Kombolcha zones, northeast Ethiopia, among pregnant women. The study involved 780 pregnant women using the cluster sampling technique. A census was conducted in 20 randomly selected clusters to identify eligible pregnant women. Actual data were collected home-to-home in the community through face-to-face interviews. Data were collected by Open Data Kit (ODK) and exported to STATA 17 for analysis. A multivariable logistic regression analysis was performed, and the goodness of the model was checked by Hosmer–Lemeshow’s test statistic and rock curve. An adjusted odds ratio with a 95% confidence interval and *p*-value < 0.05 was considered statistically significant. Result: The study revealed that 49.6% of pregnant women lack autonomy and 50% lack knowledge about early postpartum contraception, with participants’ wealth index status ranging from rich (36.6%) to poor (33.2%). The study found that 75.8% of pregnant women intended to adopt early postpartum modern contraceptive methods early after childbirth. After controlling the potential confounders, mother’s age (AOR = 6.2 [2.6–14.6], birth interval (AOR = 2.5 [1.6–3.7]), have paid work (AOR = 1.9 [1.3–2.8]), health facility from home (AOR = 2.6 [1.5–4.4]), last delivery Place (AOR = 2.4 [1.1–5.7]), knowledge on (AOR = 1.5 [1.1–2.1]), and antenatal care follow-up (AOR = 1.9 [1.2–3.3]) were significant associated factors of intention to uptake early postpartum modern contraceptive methods among pregnant women. Conclusions: The study found that 75% of the participants had the intention to adopt contraceptive methods during the early postpartum period. Identified factors influencing this intention were age, birth interval, women’s employment status, area of residence, distance to health facilities, last delivery place, knowledge of early postpartum modern contraception, gravidity, and antenatal care follow-up. These findings highlight the need for targeted interventions to address these factors, framing the intended users and enabling access to early adoption of postpartum contraceptive methods.

## 1. Introduction

Early postpartum contraception is defined as the use of any one modern birth control method (condoms, pills, injectable, implants, IUCD, or permanent methods) within the first six weeks after childbirth [1,2,3]. The early postpartum period provides a window of opportunity for women to adopt modern contraceptive methods [4]. Despite this fact, in developing countries, many postpartum women fail to obtain contraceptive methods soon after birth and become pregnant again, either to space birth or to limit birth [5]. Globally, over 90% of the anticipated 303,000 annual maternal deaths occur in six developing countries (India, Nigeria, Pakistan, the Democratic Republic of the Congo, Ethiopia, and Tanzania) [6]. Sustainable Development Goals states that by 2030, no country ought to have a maternal mortality ratio above 70 per 100,000 live births [7]. Maternal mortality is a major public health issue in Africa, particularly in Sub-Saharan and East Africa [6]. East African countries like Kenya, Uganda, Tanzania, and Somalia are especially affected, requiring improved healthcare services and education to reduce deaths [7]. In Ethiopia, maternal mortality is a vital public health issue, with 412 per 100,000 live births dying as a result of complications associated with pregnancy and childbirth [8,9]. To reduce the danger of adverse maternal, perinatal, and neonatal outcomes, the World Health Organization recommends an interval of a minimum of 24–36 months between delivery and later gestation [10].

The healthcare system in Ethiopia is a tiered structure consisting of primary, secondary, and tertiary care services aimed at improving public health, but it faces challenges such as limited resources, a shortage of healthcare professionals, and varying access across urban and rural areas. While efforts are being made to expand healthcare access and improve maternal and child health services, many rural populations still struggle with inadequate healthcare infrastructure and resources. Family planning is an integral component of maternal and child health (MCH) services in the Ethiopian health system. However, the unmet need for contraceptives during the early postpartum period in Ethiopia is unacceptably high [9]. This exposes the mothers to a high risk of unwanted pregnancies and unsafe abortions. Evidence shows that short birth intervals increase the danger of maternal, newborn, neonatal, and associated under-5 mortality; they are also related to a magnified risk of preterm birth, low birth weight, stunting, and underweight teenage mothers [10]. Early postpartum contraception may be a well-tried, cost-efficient intervention to stop each maternal and newborn death by reducing the short birth intervals, the number of abortions, and the proportion of births at high risk [11]. Despite frequent encounters of pregnant women with healthcare providers, little is currently known about the intention of pregnant women, particularly in Ethiopia, to adopt early postpartum contraceptive methods and the factors associated with such intentions [10].

Little is currently known about the intention of pregnant women, particularly in Ethiopia, to adopt early postpartum contraceptive methods and the factors associated with such intentions [12]. Nevertheless, this information is vital to the design of behavioral change strategies to increase the uptake of early postpartum contraceptive methods. Moreover, information has been scarce regarding the intention to use contraceptive methods during early postpartum [13]. In Ethiopia, 53.9% of women had resumed sexual intercourse before six weeks postpartum, whereas only 9% used a birth control method [14]. Consistent with a study on early postpartum contraceptive use and fertility intention among women in five low-income countries, 91% of women who needed to delay pregnancy for a minimum of a year nevertheless did not take birth control at six weeks postpartum [15]. 

There is a limitation in the literature regarding the intention to use contraceptives during early postpartum in Ethiopia among pregnant women. Therefore, this study aimed to fill these gaps to determine the intention to use early postpartum modern contraceptive methods and identify determinant factors among pregnant women in Kombolcha and Dessie towns in northeast Ethiopia.

This research project’s findings could have a significant contribution to local planners, programmers, nongovernmental organizations, and decision-makers to design appropriate interventions on factors influencing intention for early uptake of postpartum contraceptive methods. This could result in a steeper increase in the uptake of methods and a reduction in maternal mortality, child mortality, infant mortality, neonatal mortality, and unintended pregnancy rates.

## 2. Methods and Materials

### 2.1. Study Area, Design, and Period

The study was conducted in the Dessie and Kombolcha city zones in the Amhara regional state, northeast Ethiopia. Dessie is the administrative town of the south Wollo zone, which is situated 401 km from Addis Ababa to the north. Dessie city is split into 5 sub cities with 22 kebeles (clusters) and has 2 governmental hospitals and 10 health centers. Based on population projections for 2021, there are more than 470,000 residents, with an estimated 21,620 pregnant women in Dessie town. Kombolcha town is 30 km from Dessie city and 375 km from Addis Ababa, which is an industrial zone and dry port in northeast Ethiopia. There are more than 350,000 resident populations, with 16,100 estimated pregnant women in Kombolcha town. It is divided into 5 sub-cities with 19 kebeles and has 1 governmental hospital and 5 health centers [16]. A community-based cross-sectional study was deployed from 15 January–15 February 2023.

### 2.2. Population and Eligibility Criteria

All pregnant women between 26 and 28 weeks of gestation based on the World Health Organization pregnancy screening criteria found in Dessie and Kombolcha town zones were the source population. The study population consisted of all eligible pregnant women at 26–28 weeks of gestation found in selected clusters. At enrolment, pregnant women who reported and were confirmed by the World Health Organization (WHO) screening criteria to have a pregnancy of 26–28 weeks of gestational age were considered eligible for the study and enrolled. Pregnant women who were assumed to have serious medical problems were excluded.

### 2.3. Sample Size Determination and Procedure

The first sample size was determined to estimate a single population proportion based on the following assumptions: the proportion of pregnant women who had the intention to uptake early postpartum family planning services was assumed to be 50% [17], and the allowed margin of error was 5% with a 95% level of confidence the sample size was 384 pregnant women. By considering a cluster effect of 1.86 and a 10% non-response rate, the final sample size was 784. To check the sample size adequacy to identify factors associated with intention, we used Epi info and different factors, but it was less than that calculated by a single population proportion. Based on this, a sample size of 784 was considered for this study. A cluster sampling technique was applied to select study participants. The first 20 clusters (kebeles) were randomly selected from 41 clusters in Dessie and Kombolcha towns. A census was conducted to identify eligible pregnant women from selected clusters, and all eligible pregnant women based on The World Health Organization pregnancy screening criteria were included in the selected clusters.

### 2.4. Data Collection Tools and Procedures

Data were collected by using pre-tested interviewer-administered structured questionnaires that were adapted from different studies in the literature [8,18,19]. The questionnaire was prepared in English and then translated into the local language, Amharic, and back-translated to English by different experts to check its consistency. Then Open Data Kit (ODK) forms were prepared, validated, and aggregated through the Kobo toolbox. The ODK tool was shared with data collectors on their smartphones before data collection. When they completed daily data collection, the collected data were sent to the server data source and checked daily by the principal investigator to provide feedback immediately for the data collectors as needed.

### 2.5. Data Collection and Quality Assurance

This study was a baseline study for the next interventional RCT study to improve the uptake of early postpartum contraception. Before actual data collection, a census was conducted to register and identify eligible pregnant women from selected clusters based on local health extension workers’ support. Actual data were collected from home to home in the community following residence and contact addresses identified through the census. To ensure data quality, training was conducted for data collectors and supervisors, and a pretest was conducted to validate the tool. The tool was checked by content and face validly by senior experts. The reliability was assessed through a pretest and statistically confirmed with a Cronbach’s alpha of 0.8. The intra-variability of interviewers was tested by comparing information collected by the data collectors. When inconsistencies appeared, measures were taken, and refresher coaching for data collectors was conducted. Eight college-completed diploma nurses familiar with the local geography and fluent in the local language, Amharic, were recruited for participant recruitment and data collection. Four holders of Masters of Public Health were recruited and supervised the data collection process together with the principal investigator.

### 2.6. Data Processing and Analysis

The collected data from the open data kit (ODK) were exported to STATA for Windows version 17 for analysis. A descriptive analysis was performed by computing proportions and summary statistics. Data cleaning was performed by running frequencies, cross-tabulating, and sorting among reported variables. A backward stepwise multivariable logistic regression analysis was used to show factors determining the intention to uptake early postpartum contraceptive methods. Variables that had a *p*-value of 0.25 or less in the binary logistic regression were included in the multivariable logistic regression. Before the final model, the multicollinearity test was checked using standard error. The goodness of the model was checked by Hosmer–Lemeshow’s test statistic and the ROC (Receiver Operating Characteristic) curve. An adjusted odds ratio with a 95% confidence interval was considered to determine the strength and direction of the association. Finally, a *p*-value < 0.05 was considered statistically significant for all independent variables in the multivariable logistic regression.

### 2.7. Operational Definitions

Early postpartum contraceptive method use: is defined as the use of any modern birth control method within the first six weeks after childbirth [10].

Intention to uptake the early postpartum modern contraceptive method: Intention to use modern contraception refers to all pregnant women reporting the intent to use any of the modern contraceptive methods within 45 days after childbirth. Early postpartum contraception and birth spacing intention-related 12 questions were asked to the study participant. Then, during analysis, we used principal component analysis, and index measurement was developed. Finally, a binary dependent variable indicating an intention to use any method of modern contraception during the early postpartum period was coded as “1” and had no intention was coded as “0” based on the median score value [12,18,20].

Knowledge: We used 12 items for the early postpartum contraception knowledge-related questions, and the principal component analysis method was used with a fixed number of factors to measure knowledge. Those who had the median and the above value were taken as having good knowledge [13].

Wealth Index: We used 19 items to measure the urban wealth index, and the principal component analysis method was used with a fixed number of factors to measure wealth status. The wealth index was categorized as poor, middle income, and rich based on the percentile values [19].

Women’s autonomy: We used 23 items considering the three categories of decision-making autonomy, movement autonomy, and financial autonomy in the context of developing countries. The principal component analysis was used to develop a women’s autonomy index with a fixed number of factors to measure women’s autonomy. Those who had a value equal to the median or above were taken as autonomous and below the median value were considered not autonomous [20].

### 2.8. Ethical Issues and Approval

This study was carried out in line with the Helsinki Declaration. The actual data collection was carried out after receiving ethical approval from Jimma University, institute of health ethical review board (Reference Number JUIH/IRB 229/22). Written permission was given to all relevant authorities in the Dessie and Kombolcha town zones. Then, the participants were informed about the aim and purpose of the study, the importance of their participation, and their rights, and informed consent was obtained from the study subjects. Participants were offered a chance to withdraw from the study, and participation was entirely voluntary. Privacy and confidentiality were maintained during data collection.

## 3. Result

### 3.1. Socio-Demographic and Economic Characteristics of Participants

Out of the total 784 participants, 780 pregnant women participated in the study, resulting in an impressive response rate of 99.4%. The average age of the participants was 29 years, with a standard deviation of ±4. The mean family size was 4, with a standard deviation of ±1, and the mean age of marriage was 21 years, with a standard deviation of ±3. The majority of the study participants 730 (93.6%) were married, and 726 (93.1%) participants were from male-headed households. Approximately half of the participants, 390 (50.0%), resided in Dessie, and the majority of them, 493 (63.2%), were Muslim. Regarding employment status, only 232 (29.7%) participants had paid work and 266 (34.1%) had social health insurance (Table 1). 

### 3.2. Women’s Autonomy

The study revealed that among the study participants, 387 (49.6%) had no autonomy to decide on health, family-related issues, and resource issues (Table 2).

### 3.3. Past Obstetric Experience

The research findings indicate that 522 (66.9%) study participants have experienced 2–3 pregnancies throughout their lives and 224 (28.7%) participants reported a history of abortion. The study also showed that 128 (16.4%) participants had experienced stillbirths, and nearly half of the 414 (48.2%) participants had birth-to-pregnancy intervals of less than 24 months. Among the participants, 82 (12.1%) individuals had not undergone at least one antenatal care (ANC) follow-up, while the majority, 426 (71.6%) participants, received their ANC at the public health center. Additionally, the study participants collectively perceived that the health facility was located at a considerable distance from their homes (Table 3).

### 3.4. Knowledge and Previous Experience of Early Postpartum Modern Contraceptive Method

Among the study participants, 455 (58.4%) indicated that pregnancy could occur starting 45 days after childbirth if sexual activity is resumed, while 338 (43.3%) participants asserted that pregnancy cannot happen without the occurrence of menstruation after childbirth. The study also brought to light that 318 (40.8%) participants reported a lack of awareness that women could initiate birth control methods immediately after giving birth. In terms of awareness, 333 (42.7%) respondents stated that they had not heard about early postpartum family planning, and 160 (20.5%) participants lacked knowledge about any contraceptive methods. The overall knowledge assessment revealed that 390 (50%) participants possessed good knowledge regarding early postpartum family planning. Additionally, among the participants, 339 (43.5%) had not used contraceptive methods, and 222 (29.1%) sought contraceptive methods within 6 weeks after childbirth before the current pregnancy (Table 4).

As per the responses from study participants, only 43 (4.5%) of pregnant women express a desire for another child soon after giving birth, while 127 (30.9%) wish to have another child after the current pregnancy, though the specific timing remains undecided. Additionally, 283 (37.2%) preferred to space their pregnancies and defer the decision for a later time, while 16.5% had not yet determined whether to have another child after the current pregnancy. In general, the study revealed that 83 (10.9%) and 638 (84.6%) of pregnant women expressed a demand for early postpartum modern contraceptives to limit and space pregnancies, respectively (Figure 1).

### 3.5. Intention to Uptake Early Postpartum Modern Contraceptive Methods among Pregnant Women

From the study, it was observed that 466 (59.7%) had conversations about early postpartum family planning with their partners. The findings from the study also indicated that 119 (15.3%) participants expressed the belief that early postpartum contraception does not offer any health benefits to women, while 314 (40.3%) participants held the view that early postpartum modern contraception does not provide any health benefits to children. Moreover, a substantial proportion of participants, 661 (84.7%), believed that a large family size negatively impacts economic conditions, whereas 314 (40.3%) participants indicated that having a large family size earns respect from their husbands. Among study participants, 591 (75.6%), with a 95% confidence interval of 73–79%, had intentions to use modern contraceptives within 45 days after delivery (Table 5).

### 3.6. Factors Associated with Intention to Uptake Early Postpartum Modern Contraceptive Method

In the bivariable logistic regression analysis, respondents’ age, birth interval, women having paid work, area of residence, health facility from home, last delivery place, wealth index, knowledge on early postpartum modern contraception, gravidity, having ANC follow-up, women autonomy, and wealth index were candidate variables for multivariable logistic regression at *p*-value < 0.25. After controlling the potential confounders, age, birth interval, women having paid work, area of residence, health facility from home, last delivery place, knowledge of early postpartum modern contraception, gravidity, and having antenatal care follow-up were significant associated factors at a *p*-value < 0.05 with the intention to uptake early postpartum modern contraceptive methods among pregnant women (Table 6).

The odds of intention to uptake early postpartum modern contraceptive methods among pregnant women who live in Kombolcha town were two times higher than those who lived in Dessie town (AOR = 2.1 [1.4–3.1]). The odds of intention to uptake early postpartum modern contraceptive methods among mothers who had good knowledge of early postpartum contraception were 1.5 times higher than those mothers who had poor knowledge (AOR = 1.5 [1.1–2.1]). The odds of intention to uptake early postpartum modern contraceptive methods among women aged 19–24 were 6 times higher, 25–29 years of age were 2.2 times higher, and 30–34 years of age were 1.7 times higher compared to mothers aged more than 35 years of age (AOR = 6.2 [2.6–14.6], AOR = 2.2 [1.2–3.8], and AOR = 1.7 [1.1–2.9], respectively). Concerning the distance of a health facility from home, the odds of intention to uptake early postpartum modern contraceptive methods among women who perceived a near or medium distance between a health facility and their home were 3 and 2.6 times higher than those who said the health facility was far from their home (AOR = 3.0 [1.6–5.5] and AOR = 2.6 [1.5–4.4]), respectively. In addition, the odds of intention to uptake early postpartum modern contraceptive methods among women who had antenatal care follow-up were two times higher than those of those who did not have ANC follow-up history (AOR = 1.9 [1.1–3.3]).

Regarding the place of last delivery, women who had given birth before in a public health center were 2.4 times more likely to uptake early postpartum modern contraceptive methods as compared to those who gave birth at private health facilities (AOR = 2.4 [1.1–5.8]). Moreover, women who had paid work were two times more likely to have the intention to uptake early postpartum modern contraceptive methods than women who did not have paid work (AOR = 1.9 [1.3–2.8]). The odds of intention to uptake early postpartum modern contraceptive methods among women who had birth-to-pregnancy intervals of 24 months and above were 2.5 times higher compared to those women with a birth-to-pregnancy interval of less than 24 months (AOR = 2.5 [1.6–3.7]).

## 4. Discussion

The study finding shows a difference between the two towns, as in Kombolcha, the intention to uptake was 82.8%, and in Dessie, it was only 68.7%. The difference may be that Kombolcha town is the industrial zone and dry port of northeast Ethiopia, which means different ethnic groups and young industry workers may have more intention of birth spacing and the early adoption of postpartum contraception. The results of this study indicate the prevalence of intention to use early postpartum modern contraceptives among pregnant women was 75.6%, with a 95% CI ranging from 73% to 79%. This figure is less than the study finding regarding Tigray, which reported that 84% of the women surveyed were willing to adopt a family planning method after birth [12]. Variations may be influenced by cultural, socioeconomic, and healthcare system factors, and understanding these differences is valuable in modifying interventions. This result was slightly higher than the findings in Nigeria and Ambo, which reported 54% [17,18,21]. This variation could be explained by the difference in study setting, access to information, and availability of services used.

The findings from the study suggest that 84.5% of the participants indicated a preference for spacing pregnancies, while 11% expressed a desire to limit the number of children they have. This finding is consistent with other studies [22]. Cultural, socioeconomic, and demographic factors may influence these preferences, and understanding these variations can inform targeted healthcare strategies.

The finding from the study indicates that 56.5% of participants reported using a contraceptive method before the current pregnancy. This finding is higher than previous studies in Nigeria, which reported 39.5%, and Turkey, which reported 50% [23,24]. However, it is lower than the study performed in Ghana [22]. Cultural, socioeconomic, and demographic factors may influence these preferences, and understanding these variations can inform targeted healthcare strategies.

The results indicated that the odds for the intention to use early postpartum contraceptive methods were unexpectedly lower among women aged 35 years and above, compared to those in the 19–34 years age categories. This is consistent with other studies performed in southwest Ethiopia [25]. The possible explanations could be differences in reproductive goals, perceived fertility, previous contraceptive experiences, or cultural norms related to family planning among women in different age groups. For example, women in the 35 and above age group might have completed their desired family size or may be less concerned about postpartum contraception due to perceived lower fertility.

The findings revealed that the odds for the intention to use early postpartum contraceptive methods showed no difference based on wealth index status and women’s autonomy. The result implies that, in this study, the level of decision-making power women had within the household did not significantly influence their intention to use early postpartum contraceptives. However wealth status and social class have shown an association in another study [26]. Cultural and contextual factors may contribute to this result. The lack of differences based on wealth index and women’s autonomy emphasizes the need for a holistic approach in reproductive health programs.

The study findings indicated that good knowledge of early postpartum contraception is associated with higher odds of intention to use modern contraceptive methods, highlighting the pivotal role of information and education in shaping family planning decisions. It is consistent with other studies performed in Ambo (Ethiopia) and Nigeria [18,27]. It provides valuable insights for the development of targeted interventions and emphasizes the significance of well-informed healthcare practices in promoting reproductive health.

This finding suggests a significant association between the perceived distance to health facilities and the intention to uptake early postpartum modern contraceptive methods. The odds of intention to use early postpartum modern contraceptives are notably 3 times higher among women who perceive the health facility to be near to their home. The finding is consistent with other studies [28,29]. The result emphasizes the importance of local infrastructure and community-level healthcare services. Stakeholders can also work towards mitigating the impact of perceived distance on the intention to uptake early postpartum modern contraceptive methods through the integration of telehealth services and remote consultations to provide family planning counseling and support.

The results indicate that the odds of intention to uptake early postpartum modern contraceptive methods were 2 times higher among women who had antenatal care (ANC) follow-up compared to those who did not have ANC follow-up history, which suggests a significant association between antenatal care utilization and reproductive health intentions. This result is consistent with other studies [13,15]. Antenatal care visits provide opportunities for healthcare providers to engage with pregnant women and provide information on various aspects of reproductive health, including family planning.

The finding that women who gave birth before in a public health facility were 2.4 times more likely to uptake early postpartum modern contraceptive methods compared to those who gave birth at private health facilities is an interesting observation that may have various implications. The finding is consistent with a previous study [30]. It indicates that healthcare providers in public health centers may be more proactive in discussing family planning options during postpartum care.

The result indicating that women who had paid work were 2 times more likely to have the intention to uptake early postpartum modern contraceptive methods compared to women who did not have paid work is an interesting finding with potential implications for understanding the intersection of economic empowerment and early postpartum contraceptive intention. It is consistent with other studies [31,32]. Policymakers and program developers in reproductive health should integrate women’s empowerment and income generation activities into existing programs. 

The finding that the odds of intention to uptake early postpartum modern contraceptive methods were 2.5 times higher among women with a birth-to-pregnancy interval of 24 months and above, compared to those with an interval of less than 24 months, indicates a significant association between birth spacing and contraceptive intentions. The finding is associated with the previous literature [31,33]. It emphasizes the potential benefits of promoting birth spacing for shaping intentions to uptake early postpartum contraception.

### Limitations of the Study

The study was community-based with a larger sample size, but we elicited the information via self-reporting from respondents who were prone to social desirability bias due to the sensitive nature of the issue in the community. This study did not show the actual users among the intended users, which necessitates a follow-up study.

## 5. Conclusions

The study found that more than seven in ten of the participants expressed an intention to use early postpartum contraceptive methods. Identified factors associated with the uptake of contraceptive methods during the early postpartum period were age, birth interval, women’s employment status, area of residence, distance to health facilities, last delivery place, knowledge of early postpartum modern contraception, gravidity, and antenatal care follow-up. These findings highlight the need for targeted interventions to address these factors and promote informed decision-making regarding early postpartum contraception.

## 6. Recommendations

The recommendations aim to address the identified factors influencing the intention to use early postpartum contraceptive methods. Healthcare providers and programmers should develop and implement diverse information, education, and communication programs targeting pregnant women and women with prior experiences of narrow birth intervals focusing on enhancing knowledge about early postpartum contraceptive methods to increase the intention to use it early after childbirth. Policymakers and program developers in reproductive health should integrate women’s empowerment and income generation activities into existing programs.

Private health facilities should prioritize comprehensive counseling and education on the early uptake of contraceptive methods during the delivery process. Healthcare providers, policymakers, and advocacy groups can work collaboratively to understand and overcome the barriers influencing the lower intention to use early postpartum contraceptive methods among women aged 35 years and above. Stakeholders can work toward mitigating the impact of perceived distance on the intention to uptake early postpartum modern contraceptive methods through the integration of telehealth services and remote consultations to provide family planning counseling and support. There is a need for future studies to follow up on these women and find out what happened after their delivery.

## Figures and Tables

**Figure 1 ijerph-21-01418-f001:**
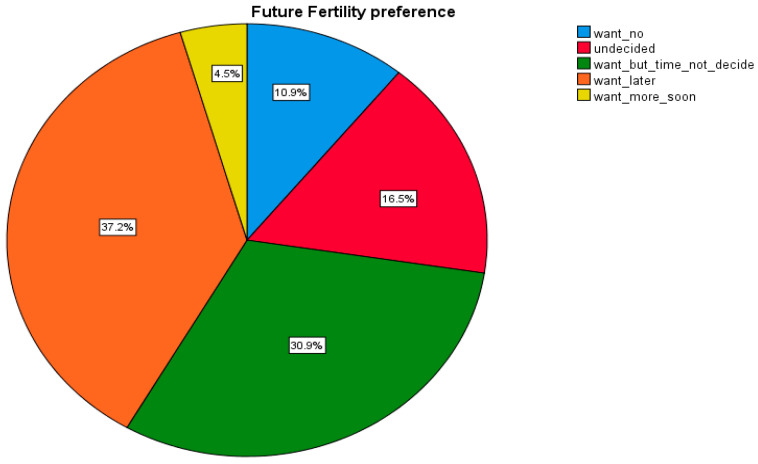
Future fertility preferences among pregnant in Dessie and Kombolcha city administration, Amhara Region, Northeast Ethiopia (n = 780).

**Table 1 ijerph-21-01418-t001:** Socio-demographic characteristics of study participants in Dessie and Kombolcha city administration, Amhara Region, Northeast Ethiopia, 2023 (n = 780).

Variables	Categories	Intention to Uptake the EPP Contraceptive Method		
		No [No (%)]	Yes [No (%)]	Total [No (%)]
Age	18–24	54 (6.9)	42 (5.4)	96 (12.3)
	25–29	161 (20.6)	134 (17.1)	295 (37.8)
	30–34	148 (19.0)	118 (15.1)	266 (34.1)
	35+	75 (9.6)	48 (6.1)	123 (15.8)
	Total	438 (56.2)	342 (43.6)	780 (100.0)
Residence	Dessie	221 (28.3)	169 (21.6)	390 (50.0)
	Kombolcha	217 (27.8)	173 (22.1)	390 (50.0)
Head of HH	Female-headed	29 (3.7)	25 (3.2)	54 (6.9)
	Male headed	409 (52.4)	317 (40.4)	726 (93.1)
Religion	Muslim	278 (35.6)	215 (27.4)	493 (63.2)
	Christian	160 (20.5)	127 (16.2)	287 (36.8)
Marital status	Married	412 (52.8)	318 (40.6)	730 (93.6)
	Separated	13 (1.7)	14 (1.8)	27 (3.5)
	Others	14 (1.8)	10 (1.3)	24 (3.1)
Women education status	Read and write	29 (3.7)	15 (1.9)	44 (5.6)
	Primary	156 (20.0)	121 (15.4)	277 (35.5)
	secondary	185 (23.7)	137 (17.5)	322 (41.3)
	College and above	68 (8.7)	69 (8.8)	137 (17.6)
Women have paid work	no	299 (38.3)	249 (31.8)	548 (70.3)
	Yes	139 (17.8)	93 (11.9)	232 (29.7)
Marital age	15–17 years	61 (7.8)	50 (6.4)	111 (14.2)
	18–24 years	314 (40.3)	254 (32.4)	568 (71.8)
	≥25 years	63 (8.1)	38 (4.8)	101 (12.9)
Family size	≥4	313 (40.1)	239 (30.5)	552 (70.8)
	>4	125 (16.0)	103 (13.1)	228 (29.2)
Wealth Index	Rich	77 (9.8)	209 (26.8)	286 (36.6)
	Middle	40 (5.1)	195 (25)	235 (30.2)
	Poor	72 (9.2)	187 (23.9)	259 (33.2)
Have health insurance	No	292 (37.4)	222 (28.3)	514 (65.9)
	Yes	146 (18.7)	120 (15.3)	266 (34.1)

**Table 2 ijerph-21-01418-t002:** Women’s autonomy status of study participants in Dessie and Kombolcha city administration, Amhara Region, Northeast Ethiopia, 2023 (n = 780).

Variables	Categories	Intention to Uptake EPP Contraceptive		
		No [No (%)]	Yes [No (%)]	Total [No (%)]
Decide on the number of children	No	86 (11.0)	53 (6.8)	139 (17.8)
	Yes	352 (45.1)	289 (37.1)	641 (82.2)
Decide on family size	No	90 (11.5)	61 (7.8)	151 (19.4)
	Yes	348 (44.6)	281 (36.0)	629 (80.6)
Leadership Group	No	84 (10.8)	64 (8.2)	148 (19.0)
	Yes	354 (45.4)	278 (35.6)	632 (81.0)
Decied health facility visit	No	82 (10.5)	61 (7.8)	143 (18.3)
	Yes	356 (45.6)	281 (36.0)	637 (81.7)
Take a child health facility	No	79 (10.1)	62 (7.9)	141 (18.1)
	Yes	359 (46.0)	280 (35.9)	639 (81.9)
Decied future fertility	No	86 (11.0)	53 (6.8)	139 (17.8)
	Yes	352 (45.1)	289 (37.1)	641 (82.2)
Right to use money for health	No	84 (10.8)	64 (8.2)	148 (19.0)
	Yes	354 (45.4)	278 (35.6)	632 (81.0)
Decided to use modern FP methods	No	82 (10.5)	61 (7.8)	143 (18.3)
	Yes	356 (45.6)	281 (36.0)	637 (81.7)
Decied desired fertility	No	79 (10.1)	62 (7.9)	141 (18.1)
	Yes	359 (46.0)	280 (35.9)	639 (81.9)
Decied income earning	No	86 (11.0)	53 (6.8)	139 (17.8)
	Yes	352 (45.1)	289 (37.1)	641 (82.2)
Decied purchasing	No	90 (11.5)	61 (7.8)	151 (19.4)
	Yes	348 (44.6)	281 (36.0)	629 (80.6)
Decied expenditure on health	No	84 (10.8)	64 (8.2)	148 (19.0)
	Yes	354 (45.4)	278 (35.6)	632 (81.0)
Decied expenditure on education	No	82 (10.5)	61 (7.8)	143 (18.3)
	Yes	356 (45.6)	281 (36.0)	637 (81.7)
Decied expenditure on nutrition	No	79 (10.1)	62 (7.9)	141 (18.1)
	Yes	359 (46.0)	280 (35.9)	639 (81.9)
Control over house	No	90 (11.5)	61 (7.8)	151 (19.4)
	Yes	348 (44.6)	281 (36.0)	629 (80.6)
Control paid work	No	82 (10.5)	61 (7.8)	143 (18.3)
	Yes	356 (45.6)	281 (36.0)	637 (81.7)

**Table 3 ijerph-21-01418-t003:** Past obstetric experiences of study participants in Dessie and Kombolcha city administration, Amhara Region, Northeast Ethiopia, 2023 (n = 780).

Variables	Categories	Frequency	Percent (%)
Number of pregnancies ever	1	103	13.2
	2–3	522	66.9
	≥4	155	19.9
Had abortion history	no	556	71.3
	Yes	224	28.7
Had stillbirth history	no	652	83.6
	yes	128	16.4
Birth to pregnancy interval	<12 months	184	23.6
	12–24 months	192	24.6
	24–36 months	222	28.5
	>36 months	182	23.3
Had ANC for last delivery	no	82	12.1
	yes	595	87.8
Gestational age during health facility visit to ANC	<16 weeks	319	53.5
	16–28 weeks	215	36.1
	28–32 weeks	23	3.8
	32–36 weeks	38	6.4
Number of ANC visits to previous pregnancy	once	92	15.5
	twice	112	18.8
	three	164	27.6
	Four	227	38.2
ANC place to last pregnancy	Health center	426	71.6
	Hospital	79	13.3
	Private health facility	90	15.1
Health facility from home	Good near	164	21.0
	Medium	523	67.1
	Too far	93	11.9
Place of last delivery	Health center	366	61.5
	Hospital	150	25.2
	Private health facility	71	11.9
	Home	8	1.3

**Table 4 ijerph-21-01418-t004:** Knowledge and previous experiences on early postpartum modern contraceptive methods of study participants in Dessie and Kombolcha city administration, Amhara Region, Northeast Ethiopia (n = 780).

Variables	Categories	Intention to Uptake EPP Contraception		
		No [No (%)]	Yes [No (%)]	Total [No (%)]
Time be pregnant after childbirth if she has sexual practice	After 45 days	243 (31.2)	212 (27.2)	455 (58.4)
	After 2 years	28 (3.6)	7 (0.9)	35 (4.5)
	After 6 months	128 (16.4)	105 (13.5)	233 (29.9)
	Do not know	39 (5.0)	18 (2.3)	57 (7.3)
Pregnancy can happen without seeing menstruation after childbirth	no	198 (25.4)	140 (17.9)	338 (43.3)
	yes	240 (30.8)	202 (25.9)	442 (56.7)
A narrow birth interval has a negative health impact on the mother	no	44 (5.6)	38 (4.9)	82 (10.5)
	yes	394 (50.5)	304 (39.0)	698 (89.5)
A narrow birth interval has a negative health impact on children	with no	44 (5.6)	38 (4.9)	82 (10.5)
	yes	394 (50.5)	304 (39.0)	698 (89.5)
A woman can take birth control method immediately after birth	no	190 (24.4)	128 (16.4)	318 (40.8)
	yes	248 (31.8)	214 (27.4)	462 (59.2)
Minimum birth interval to the health of the mother and the child	1 year	34 (4.4)	18 (2.3)	52 (6.7)
	2–3 years	220 (28.2)	169 (21.7)	389 (49.9)
	4–5 years	151 (19.4)	139 (17.8)	290 (37.2)
	Do not know	33 (4.2)	16 (2.1)	49 (6.3)
Heard about early postpartum family planning	no	208 (26.7)	125 (16.0)	333 (42.7)
	yes	230 (29.5)	217 (27.8)	447 (57.3)
Know any contraceptive method	no	91 (11.7)	69 (8.8)	160 (20.5)
	yes	347 (44.5)	273 (35.0)	620 (79.5)
Pregnancy can happen to start 45 days after childbirth if the woman has sexual practice	no	119 (15.3)	0 (0.0)	119 (15.3)
	yes	319 (40.9)	342 (43.8)	661 (84.7)
Used contraceptive method before this pregnancy	No	85 (11.1)	248 (32.5)	333 (43.5%)
	Yes	103 (13.5)	328 (42.9)	431 (56.5)
Early postpartum contraception before this pregnancy	No	136 (17.8 (	406 (53.1)	542 (70.9)
	Yes	52 (6.8)	170 (22.5)	222 (29.1)

Future fertility preference.

**Table 5 ijerph-21-01418-t005:** Intention to uptake early postpartum modern contraceptive methods among pregnant women in Dessie and Kombolcha city administration, Amhara Region, Northeast Ethiopia, 2023 (n = 780).

Variables	Categories	Number	Percent (%)
Pregnancy can happen starting from 45 days after childbirth if the woman has sexual practice.	no	119	15.3
	yes	661	84.7
Discussion on early postpartum family planning with one’s partner.	no	314	40.3
	yes	466	59.7
Discussion on early postpartum family planning with family.	no	316	40.5
	yes	464	59.5
Discussion on early postpartum family planning with friends	no	415	53.2
	yes	365	46.8
EPP contraception has benefits to the health of women	no	119	15.3
	yes	661	84.7
EPP contraception has benefits for the health child	no	314	40.3
	yes	466	59.7
Large family size affects economic conditions negatively	no	119	15.3
	yes	661	84.7
Large family size earns respect from the husband	no	314	40.3
	yes	466	59.7
Large family size earns the respect of the community	no	119	15.3
	yes	661	84.7
Large family size earns respect from a religious leader	no	314	40.3
	yes	466	59.7
Large family size affects maternal and child health	no	314	40.3
	yes	466	59.7
Will take modern contraceptive method within 45 days after delivery	no	189	24.2
	yes	591	75.8

**Table 6 ijerph-21-01418-t006:** Bivariable and multiple logistic regression on factors associated with intention to uptake early postpartum contraceptive method among pregnant women in Dessie and Kombolcha Zones, North-East Ethiopia (n = 780).

Variables	Category	Intention to Uptake EPP Contraceptive		COR (95% CI)	AOR (95% CI)
		No	Yes		
Residence	Kombolcha	67	323	2.2 [1.5–3.0] *	2.1 [1.4–3.1] **
	Dessie	122	268	1	1
Have paid work	no	93	139	1	1
	yes	96	452	3.2 [2.2–4.4] *	1.9 [1.3–2.8] **
Health facility from home	near to home	33	131	3.5 [2.0–6.2] *	3.0 [1.5–5.5] **
	medium	112	411	3.2 [2.0–5.2] *	2.6 [1.5–4.4] **
	Far	44	49	1	1
Had ANC follow-up	no	66	119	1	1
	yes	123	472	2.1 [1.4–3.0] *	1.9 [1.1–3.3] *
Place of last delivery	health center	89	336	2.7 [1.6–4.6] *	2.4 [1.0–5.7] *
	hospital	30	120	2.9 [1.5–5.4] *	1.2 [0.1–9.7]
	private	30	41	1	1
Age	18–24	12	84	2.8 [1.4–5.9] *	6.2 [2.6–14.6] **
	25–29	76	219	1.2 [0.7–1.9]	2.2 [1.2–3.8] **
	30–34	65	201	1.3 [0.7–2.0]	1.7 [1.0–2.9] *
	35+	36	87	1	1
Knowledge about EPP contraception	Good	76	314	1.7 [1.2–2.3] *	1.5 [1.1–2.1] *
	Poor	113	277	1	1
Pregnancy status	Primi gravida	35	68	1	1
	Multi gravida	154	523	1.7 [1.1–2.7] *	1.6 [0.7–3.3]
Birth interval	<24 months	127	249	1	1
	>24 months	62	342	2.8 [1.9–3.9] *	2.5 [1.6–3.7] **
Wealth Index	Poor	77	209	1.1 [0.7–1.5]	0.8 [0.5–1.3]
	Middle	40	195	1.8 [1.2–2.9] *	1.4 [0.8–2.3]
	Rich	72	187	1	1

COR:—Crude Odds Ratio, AOR:—Adjusted Odds Ratio, 1:—Reference, *****—*p*-value < 0.05, **—*p*-value < 0.01.

## Data Availability

All data underlying the findings described in the manuscript are freely available to other researchers within the manuscript itself, and all raw data required to replicate the results of the study were uploaded.

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
