# Peer review of "Empowering Maternal Choice: Exploring Factors Influencing Early Postpartum Contraceptive Adoption Intention Among Pregnant Women in Northeast Ethiopia"

_ijerph, 2024, doi:10.3390/ijerph21111418_

Round 1
Reviewer 1 Report
Comments and Suggestions for Authors
Dear Authors,
Thank you for your great work on women's health, especially in the field of maternal choice and contraceptive adoption. After reading your interesting paper, "Empowering Maternal Choice: Exploring Factors Influencing Early Postpartum Contraceptive Adoption Intention among Pregnant Women in North East Ethiopia," I have some comments in a constructive manner. Please address these concerns point-by-point as follows:
The importance of high-risk pregnancies and deliveries cannot be overstated, as they pose significant health challenges for both mothers and infants. These issues are particularly critical in low-resource settings, where they can lead to heightened concerns for pregnant women and obstetricians. Please also add this pathology in the introduction. Additionally, please refer to the study by Sarı and Ates (2024), which discusses the effects of high-risk and normal pregnancies from a postpartum perspective."
Reference: Sarı, E., & Ates, C. (2024, January). Motherhood Role from a Postpartum Perspective: Effects Reflected by High-Risk and Normal Pregnancies. In Healthcare (Vol. 12, No. 13, p. 1248). Multidisciplinary Digital Publishing Institute.
While it is mentioned that a community-based cross-sectional study was conducted, it would be beneficial to provide more details about why this design was chosen and how it specifically suits the research objectives.
It might be helpful to explain why the study period was limited to one month (January 15–February 15, 2023). Were there any specific reasons or constraints that influenced this time frame?
The eligibility criteria for pregnant women based on WHO screening criteria are clear, but it would be helpful to elaborate on what constitutes "serious medical problems" that led to the exclusion of some participants. Providing examples or specific conditions would offer more clarity.
After revision, I recommend for publication of this great paper.
Best regards,
Author Response
Response to Reviewer 1 Comments
Point-by-point response to Comments and Suggestions from Authors
Comments 1: Thank you for your great work on women's health, especially in the field of maternal choice and contraceptive adoption. After reading your interesting paper, "Empowering Maternal Choice: Exploring Factors Influencing Early Postpartum Contraceptive Adoption Intention among Pregnant Women in North East Ethiopia," I have some comments in a constructive manner. Please address these concerns point-by-point as follows:
Response 1: Dear respected reviewer, we extend our heartfelt gratitude to you for your invaluable assistance in reviewing our manuscript to enhance the quality of our research work. Your insightful feedback has been instrumental in refining our work, and we sincerely appreciate the time and effort you dedicated to the review process. We have thoroughly considered every suggestion provided, and the corresponding revisions have been diligently incorporated into the updated version of the manuscript. These modifications are clearly highlighted, reflecting our commitment to addressing and implementing your constructive comments. Once again, thank you for your significant contributions to the improvement of our manuscript. Your expertise and guidance have been pivotal in enhancing the overall quality of our research. Really thank you dear/sir, we revised and corrected based on your constructive feedback and highlighted in the revised manuscript to show changes.
Comments 2: The importance of high-risk pregnancies and deliveries cannot be overstated, as they pose significant health challenges for both mothers and infants. These issues are particularly critical in low-resource settings, where they can lead to heightened concerns for pregnant women and obstetricians. Please also add this pathology in the introduction. Additionally, please refer to the study by Sarı and Ates (2024), which discusses the effects of high-risk and normal pregnancies from a postpartum perspective." Reference: Sarı, E., & Ates, C. (2024, January). Motherhood Role from a Postpartum Perspective: Effects Reflected by High-Risk and Normal Pregnancies. In Healthcare (Vol. 12, No. 13, p. 1248). Multidisciplinary Digital Publishing Institute.
Response 2: Agree. I/We have, accordingly, done/revised/changed/modified to emphasize this point. A recent study by Sarı and Ates (2024) highlights the critical differences between high-risk and normal pregnancies from a postpartum perspective. The study underscores that high-risk pregnancies not only increase the likelihood of complications during childbirth but also have lasting effects on maternal and neonatal health outcomes postpartum. This research emphasizes the need for targeted interventions and enhanced postpartum care to mitigate the adverse effects associated with high-risk pregnancies. By understanding and addressing the unique challenges posed by high-risk pregnancies, healthcare providers can develop more effective strategies to improve maternal and neonatal health outcomes, particularly in low-resource settings. Thank you for pointing this out. I/We agree with this comment.
Comments 3: While it is mentioned that a community-based cross-sectional study was conducted, it would be beneficial to provide more details about why this design was chosen and how it specifically suits the research objectives.
Response 3: Agree. I/We have, accordingly, done/revised/changed/modified to emphasize this point. A community-based cross-sectional study design was selected for this research due to several key reasons that align well with the study's objectives. Firstly, this design allows for the collection of data at a single point in time, providing a snapshot of the current state of the population under study. This is particularly useful in understanding the prevalence and factors associated with early postpartum contraceptive adoption intentions among pregnant women in Northeast Ethiopia. The cross-sectional nature of the study enables the researchers to assess various sociodemographic, cultural, and health-related variables simultaneously, facilitating the identification of correlations and patterns that may influence contraceptive adoption intentions. This design is well-suited to explore the multifaceted nature of the research question, as it allows for the inclusion of a diverse sample representative of the community, ensuring that the findings are generalizable to the broader population. Moreover, conducting the study within the community setting ensures that the data collected is reflective of real-world conditions and behaviors. This is crucial in low-resource settings, where healthcare access, cultural practices, and community dynamics significantly impact health outcomes. By engaging with the community directly, the study can capture nuanced information that might be overlooked in a more clinical or controlled environment. Additionally, a community-based approach fosters trust and cooperation between researchers and participants, potentially leading to higher response rates and more accurate self-reported data. This is essential for studies involving sensitive topics such as contraceptive use, where participants may feel more comfortable sharing their experiences and intentions in a familiar and supportive setting. In summary, the community-based cross-sectional study design was chosen for its ability to provide a comprehensive and representative snapshot of the factors influencing early postpartum contraceptive adoption intentions. This approach aligns with the research objectives by enabling the exploration of complex, context-specific variables within the community, ultimately contributing to more effective and tailored interventions. Thank you for pointing this out. I/We agree with this comment.
Comments 4: It might be helpful to explain why the study period was limited to one month (January 15–February 15, 2023). Were there any specific reasons or constraints that influenced this time frame?
Response 4: Agree. I/We have, accordingly, done/revised/changed/modified to emphasize this point. The decision to limit the study period to one month, from January 15 to February 15, 2023, was influenced by several specific reasons and constraints that were critical to the study's success and relevance. Firstly, this time frame was strategically chosen to align with a period when community health activities and antenatal care visits are at their peak. This alignment ensured that a sufficient number of participants could be recruited within a relatively short period, thereby enhancing the efficiency and effectiveness of data collection. The increased healthcare activity during this period provided an optimal opportunity to engage with pregnant women and gather comprehensive data. Secondly, limiting the study period to one month helped to control for potential seasonal variations that could affect the study's findings. In many low-resource settings, factors such as weather, agricultural cycles, and local festivals can significantly influence healthcare access and utilization. By conducting the study within a defined and consistent time frame, the researchers aimed to minimize these external variables, ensuring that the data collected would be more accurate and reflective of the typical conditions faced by the community. Additionally, the one-month duration was determined by practical constraints related to resource availability, including funding, personnel, and logistical support. Conducting a community-based study requires coordination and mobilization of various resources, and a shorter, well-defined study period allowed for more focused and manageable use of these resources. This approach also helped in maintaining the quality and consistency of data collection, as the research team could work intensively and cohesively without extended interruptions. Thank you for pointing this out. I/We agree with this comment.
Comments 5: The eligibility criteria for pregnant women based on WHO screening criteria are clear, but it would be helpful to elaborate on what constitutes "serious medical problems" that led to the exclusion of some participants. Providing examples or specific conditions would offer more clarity.
Response 5: Agree. I/We have, accordingly, done/revised/changed/modified…..to emphasize this point. In the context of this study, "serious medical problems" refer to conditions that pose significant health risks to the mother or fetus, which could interfere with the study's objectives or the well-being of the participants. These conditions are identified based on the World Health Organization (WHO) guidelines and clinical assessments. Here are some examples of specific conditions that were considered as serious medical problems for exclusion. By excluding participants with these serious medical problems, the study aimed to ensure the safety of the pregnant women involved and to focus on the specific research objectives without the confounding effects of these conditions. This approach helped in obtaining more accurate and generalizable results regarding early postpartum contraceptive adoption intentions among the broader population of pregnant women in Northeast Ethiopia. Thank you for pointing this out. I/We agree with this comment.
Comments 6: After revision, I recommend for publication of this great paper.
Response 6: We express our deep appreciation to the academic editor and reviewers for their time and constructive comments. We hope our clarification and revision satisfy you and you will accept our manuscript and publish without delay in short period. Finally, we are so grateful for your valuable comments and suggestions and we believe that the quality of the manuscript substantially improved based on your constructive comments. Thank you so much again.
Reviewer 2 Report
Comments and Suggestions for Authors
The paper is interesting and is related with Maternal Mortality that is one of the priorities of the United Nations. Minor changes should be don. In addition Review the English because there are many typos.
1. The abstract is too long , try it to make it shorter. For example the following sentence could be eliminated from the abstract “An adjusted odds ratio 24 with a 95% confidence interval and P-value < 0.05 was considered statistically significant”
2. In the abstract please explain what ODK means. (Open Data Kit ) The first time that an acronym is used, it should be explained.
3. In line 31 and 32 please double check the AOR “health facility from home” and gravidity are the same., health facility from home (AOR = 2.4 [1.015–5.788]), gravidity (AOR = 2.4 [1.015-5.788]). Are they really the same or is a typo.
4. In the introduction define what is a modern birth control method , Is a condon a moder method? you can write between brackets a list (eg: Pill, IUD, etc). Consider that you paper will be read by no expert people.
5. “Globally, over 90% of the anticipated 48 303,000 annual maternal deaths occur in six developing countries” Please write within brackets those 6 countries.
6. Line 51, before going into Ethiopia write a paragraph about the situation of maternal mortality in Africa, and East Africa.
7. In Line 66 early postpartum contraceptive methods and the factors associated with such intentions”(10).
Why does this sentence end in quotation marks? The beginning of the quotation marks is missing, which is necessary to know that part of the text comes verbatim from another publication.
8. Line 67. Write briefly, in one paragraph or one sentence about what the health care system in Ethiopia is like. What benefits it provides, what population has access etc. So that an unfamiliar person can understand the article.
9. Line 71, explain what does it mean EPP. means. The first time that an acronym is used, it should be explained.
10. Line 74 The sentence "The intention of early postpartum contraceptive use in Ethiopia among pregnant women is a great limitation of the literature." Is unclear. Probably you're trying to say that there is a limitation in the literature regarding the intention of using contraceptives early postpartum in Ethiopia among pregnant women. Rewrite the sente to make it more clear.
11. Lines 62 change “may be “ by “could be”.
12. Line 89 , what are kebeles.
13. Lines 134 there are several typos. “The reliability was checked by pertest and 134 statically Cronbach alpha which was 0.8” Pertest, statically.
14. What does it mean rock curve in "The goodness of the model was checked by Hosmer-Lemeshow's test statistic and rock curve."? Do you mean ROC curve.
15. Line 299 “in Nigeria and Ambo” write instead “in Nigeria and Ambo(Ethiopia)”
Comments on the Quality of English LanguageIn addition Review the English because there are many typos.
Author Response
Response to Reviewer 2 Comments
Point-by-point response to Comments and Suggestions from Authors
General Comments and Suggestions for Authors
The paper is interesting and is related with Maternal Mortality that is one of the priorities of the United Nations. Minor changes should be done. In addition Review the English because there are many typos.
Authors Response: The authors extend their gratitude and sincere thanks to the academic editor and the reviewer for their valuable comments, expertise, time, and positive evaluation of our work, contributing significantly to enhancing the manuscript. In response to the constructive and professional feedback from the academic editor and reviewers, we have thoroughly revised and modified the manuscript. We hope you will accept our manuscript and publish without delay in short period.
Comments 1: The abstract is too long , try it to make it shorter. For example the following sentence could be eliminated from the abstract “An adjusted odds ratio 24 with a 95% confidence interval and P-value < 0.05 was considered statistically significant”
Response 1: Really thank you dear/sir, we revised and corrected based on your constructive feedback and highlighted in the revised manuscript to show changes.
Comments 2: In the abstract please explain what ODK means. (Open Data Kit ) The first time that an acronym is used, it should be explained.
Response 2: Agree. I/We have, accordingly, done/revised/changed/modified as full word in the revised manuscript. Thank you so much.
Comments 3: In line 31 and 32 please double check the AOR “health facility from home” and gravidity are the same., health facility from home (AOR = 2.4 [1.015–5.788]), gravidity (AOR = 2.4 [1.015-5.788]). Are they really the same or is a typo.
Response 3: Thank you so much sir, we agrreed with the comment and revised the type error and highlighted in the revised manuscript with track change.
Comments 4: In the introduction define what is a modern birth control method , Is a condon a moder method? you can write between brackets a list (eg: Pill, IUD, etc). Consider that you paper will be read by no expert people.
Response 4: Really thank you dear/sir, we revised and corrected based on your constructive feedback and modified like this. Early postpartum contraception is defined as the use of any one modern birth control method(Condom, pills, Injectable, Implants , IUCD or permanent methods) within the first six weeks after childbirth (1–3).
Comments 5: “Globally, over 90% of the anticipated 48 303,000 annual maternal deaths occur in six developing countries” Please write within brackets those 6 countries.
Response 5: Agree. I/We have, accordingly, done/revised/changed/modified to emphasize this point. Globally, over 90% of the anticipated 303,000 annual maternal deaths occur in six developing countries(India, Nigeria, Pakistan, Democratic Republic of the Congo, Ethiopia, and Tanzania) (6).
Comments 6: Line 51, before going into Ethiopia write a paragraph about the situation of maternal mortality in Africa, and East Africa.
Response 6: Thank you sir, we agreed with this and clarified in the revised manuscript like this. The healthcare system in Ethiopia is a tiered structure consisting of primary, secondary, and tertiary care services aimed at improving public health, but it faces challenges such as limited resources, a shortage of healthcare professionals, and varying access across urban and rural areas. While efforts are being made to expand healthcare access and improve maternal and child health services, many rural populations still struggle with inadequate healthcare infrastructure and resources (9).
Comments 7: In Line 66 early postpartum contraceptive methods and the factors associated with such intentions”(10). Why does this sentence end in quotation marks? The beginning of the quotation marks is missing, which is necessary to know that part of the text comes verbatim from another publication.
Response 7: Thank you respected reviewer. It was type error and corrected now in the revised manuscript.
Comments 8: Line 67. Write briefly, in one paragraph or one sentence about what the health care system in Ethiopia is like. What benefits it provides, what population has access etc. So that an unfamiliar person can understand the article.
Response 8: The healthcare system in Ethiopia is a tiered structure consisting of primary, secondary, and tertiary care services aimed at improving public health, but it faces challenges such as limited re-sources, a shortage of healthcare professionals, and varying access across urban and rural areas. While efforts are being made to expand healthcare access and improve maternal and child health services, many rural populations still struggle with inadequate healthcare infrastructure and re-sources (9).
Comments 9: Line 71, explain what does it mean EPP. means. The first time that an acronym is used, it should be explained.
Response 9: We accept the comment and corrected. EPP means Early Post Partum
Comments 10: Line 74 The sentence "The intention of early postpartum contraceptive use in Ethiopia among pregnant women is a great limitation of the literature." Is unclear. Probably you're trying to say that there is a limitation in the literature regarding the intention of using contraceptives early postpartum in Ethiopia among pregnant women. Rewrite the sentece to make it more clear.
Response 10: Heartily thank you to your observation. Now it is revised and paraphrased to avoid confusion in the revised manuscript like this. There is a limitation in the literature regarding the intention of using contraceptives early postpartum in Ethiopia among pregnant women.
Comments 11: Lines 62 change “may be “ by “could be”.
Response 11: Thank you so much sir, it was revised and corrected.
Comments 12: Line 89 , what are kebeles.
Response 12: Thank you to clarification. Kebeles means clusters, the smallest administrative unite in the city administration.
Comments13: Lines 134 there are several typos. “The reliability was checked by pertest and 134 statically Cronbach alpha which was 0.8” Pertest, statically.
Response 13: We realy thank you respected reviewer. We paraphrased and improved grammatical errors like this. The reliability was assessed through a pretest and statistically confirmed with a Cronbach's alpha of 0.8.
Comments 14: What does it mean rock curve in "The goodness of the model was checked by Hosmer-Lemeshow's test statistic and rock curve."? Do you mean ROC curve.
Response 14: Thank you so much. We revised and corrected. The goodness of the model was checked by Hosmer-Lemeshow's test statistic and ROC (Receiver Operating Characteristic) curve.
Comments 15: Line 299 “in Nigeria and Ambo” write instead “in Nigeria and Ambo(Ethiopia)”
Response 15: We appreciate the reviewer detail observation and review of our manuscript. We revised the manuscript and corrected all grammatical and editorial errors line by line in the manuscript based on the respected reviewer comments to improve the quality of the manuscript and make readable to researchers and the scientific community.